# Associations between Fasting Duration, Timing of First and Last Meal, and Cardiometabolic Endpoints in the National Health and Nutrition Examination Survey

**DOI:** 10.3390/nu13082686

**Published:** 2021-08-03

**Authors:** Michael D. Wirth, Longgang Zhao, Gabrielle M. Turner-McGrievy, Andrew Ortaglia

**Affiliations:** 1College of Nursing, University of South Carolina, 1601 Greene Street, Columbia, SC 29208, USA; 2Department of Epidemiology and Biostatistics, Arnold School of Public Health, University of South Carolina, 915 Greene Street, Columbia, SC 29208, USA; lz7@email.sc.edu (L.Z.); ortaglia@mailbox.sc.edu (A.O.); 3Department of Health Promotion, Education, and Behavior, Arnold School of Public Health, University of South Carolina, 915 Greene Street, Columbia, SC 29209, USA; brie@sc.edu

**Keywords:** chrononutrition, inflammation, metabolism, NHANES, fasting

## Abstract

Background: Research indicates potential cardiometabolic benefits of energy consumption earlier in the day. This study examined the association between fasting duration, timing of first and last meals, and cardiometabolic endpoints using data from the National Health and Nutrition Examination Survey (NHANES). Methods: Cross-sectional data from NHANES (2005–2016) were utilized. Diet was obtained from one to two 24-h dietary recalls to characterize nighttime fasting duration and timing of first and last meal. Blood samples were obtained for characterization of C-reactive protein (CRP); glycosylated hemoglobin (HbA1c %); insulin; glucose; and high-density lipoprotein (HDL), low-density lipoprotein (LDL), and total cholesterol. Survey design procedures for adjusted linear and logistic regression were performed. Results: Every one-hour increase in nighttime fasting duration was associated with a significantly higher insulin and CRP, and lower HDL. Every one-hour increase in timing of the last meal of the day was statistically significantly associated with higher HbA1c and lower LDL. Every one-hour increase in first mealtime was associated with higher CRP (β = 0.044, *p* = 0.0106), insulin (β = 0.429, *p* < 0.01), and glucose (β = 0.662, *p* < 0.01), and lower HDL (β = −0.377, *p* < 0.01). Conclusion: In this large public health dataset, evidence for the beneficial effect of starting energy consumption earlier in the day on cardiometabolic endpoints was observed.

## 1. Introduction

Diet is one of the strongest modifiers of cardiometabolic markers [1,2,3,4]. However, for much of the American adult population, dietary modification has been a challenge and has not led to significant public health improvements [5,6]. This is evidenced by the obesity and diabetes epidemics in the United States [7,8]. Furthermore, as a population, Americans have been shown to not follow dietary recommendations [9]. Attempts to lessen the obesity burden have mostly focused on reducing overall energy intake [10,11,12]. While modest caloric restriction (~500 kilocalories (kcals)/day reduction) has led to weight loss and improvements in other cardiometabolic endpoints [13,14,15], participants often experience increases in hunger [16,17,18], making it difficult to sustain this type of behavior. Furthermore, most will regain their weight-loss over a period of years [5,6].

Intermittent fasting (IF) involves varying degrees of voluntary restriction of energy for some period of time [19], and has been associated with weight loss [20,21,22]. IF covers protocols such as complete day or modified 5:2 regimens, time-restricted feeding (TRF) in which timing of food consumption is restricted to a time window (e.g., 8 h) during the day, and fasting due to religious beliefs such as Ramadan [19].

Of growing interest are protocols involving early TRF (eTRF). However, there are no precise definitions of eTRF; generally, consumption typically ends by 4 or 5 PM. Research shows the profound effect that IF can have on a range of outcomes including weight loss and metabolic disease [23,24,25]. Biologically, irrespective of weight loss, IF can elicit adaptive cellular responses within and between organs in a way that reduces inflammation, and improves glucose regulation, and resistance against oxidative and metabolic stress [23].

Recently, Pellegrini and colleagues conducted a meta-analysis of 11 clinical trials or pragmatic designs examining the impact of either TRF or eTRF on weight loss. Compared to *ad libitum* eating control groups, those undergoing TRF protocols (weighted mean difference = −1.07 kg, 95% confidence interval [95%CI] = −1.74 to −0.40) lost more weight [26]. Overall, reductions in the mean difference of fasting glucose between TRF and *ad libitum* groups were found (weighted mean difference = −1.71 mg/dL, 95%CI = −3.20 to −0.21). However, no such finding for insulin was found [26]. In pragmatic trials of Ramadan, fasting plasma glucose either showed a null change or an increase [26,27,28]. This may be related to circadian rhythms with decreased insulin secretion and peripheral and hepatic insulin resistance towards the evening hours [29,30].

Results for the effect of TRF on measures of inflammation are mixed at best. More observational or pragmatic designs tend to show an improvement in inflammatory profiles with TRF [31,32]. However, in more controlled studies, such as randomized controlled trials (RCTs), associations between TRF and inflammation tend to be null [33,34]. Lastly, the meta-analysis by Pellegrini and colleagues found null associations between TRF and cholesterol (total, high-density lipoprotein [HDL], low density lipoprotein [LDL]), and triglycerides [26], although other types of fasting, such as alternate day fasting, have shown improvements in levels of these markers, as indicated by Stekovic and colleagues, as an example [35].

A recent systematic review of IF approaches on a range of outcomes concluded that there is a large degree of heterogeneity between studies and findings [36]. One methodological flaw is that TRF naturally leads to moderate calorie restriction. This makes parsing out the effect of TRF from calorie restriction difficult. [37,38]. For example, a review found that 9 of 11 studies comparing some form of IF to continuous calorie restriction found similar effects on weight reduction. Most of those 9 studies also showed a decrease in energy intake among the IF groups [39]. Interestingly, one study compared alternate-day fasting with energy restriction to alternate-day fasting without energy restriction over three weeks. The fasting group with energy restriction had decreased total body mass and fat mass, but such reductions among the fasting group without energy restriction was not observed [40]. Regardless, much of the evidence related to TRF and cardiometabolic health have been in the form of highly controlled RCTs of short duration or pragmatic designs using religious fasting [36,37].

Given that prior TRF work may be biased by inherent calorie restriction, it was broadly theorized that usage of a population-based study, such as the National Health and Nutrition Examination Survey (NHANES), would be able to assess the association between fasting and cardiometabolic markers without overlapping calorie restriction as most people are not actively dieting or specifically cutting calories [37,38]. Additionally, a larger population, especially a population such as NHANES, leads to greater power to detect associations. NHANES also is representative of the general U.S. population, whereas smaller RCT populations usually are not.

Therefore, this study examined the impact of fasting on cardiometabolic markers using data from the NHANES. It was hypothesized that those with shorter nighttime fasting will have higher values of CRP, glycosylated hemoglobin (HbA1c %), insulin, glucose, LDL cholesterol, and total cholesterol; and lower HDL cholesterol compared to those with longer fasting durations. The impact of the timing of the last daily meal and first daily meal with the outcomes was explored.

## 2. Materials and Methods

### 2.1. Study Population

NHANES regularly collects data on American adults and children using a complex, multistage, probability sampling design making it a representative U.S.-based sample. These cross-sectional data are collected in 2-year cycles. For this analysis, NHANES data were restricted to those at least 18 years of age. Initially, participants are asked to complete a battery of questionnaires obtaining information on demographics, medical histories, socioeconomics, and behavioral and lifestyle habits. Then, participants were invited to attend a mobile examination clinic (MEC) where clinic measurements are performed. Additionally, biological samples are obtained. Full information on NHANES and the types of data available can be found elsewhere [41]. Informed consent was provided by participants and the National Center for Health Statistics Research and Ethics Review Board continually approves NHANES.

### 2.2. Outcome Assessment

While attending the MEC and in a seated position, blood was drawn by a trained phlebotomist. CRP was only available for three 2-year cycles and had an unweighted sample size of 15,939. Glucose, insulin and LDL were available for all six of the 2-year cycles but only for a subsample of the population (*n* = 15,068 for glucose, 14,757 for insulin, and 14,539 for LDL). The remaining outcomes (total cholesterol, HDL, HbA1c %) were available for all six of the 2-year cycles for most of the adult (i.e., ≥18 years) participants (*n* = 30,793 for total cholesterol; 30,793 for HDL; and 31,069 for HbA1c %). Numerous protocols underwent quality control procedures including performing blind split samples collected on “dry run” sessions within the MEC. Each lab involved in analyzing the blood samples underwent rigorous QA/QC procedures which included a random 2% retesting [41]. All outcome measures were analyzed continuously using linear regression (see statistical analysis section). Additionally, each has a standardized cut-point used in clinical practice. The cut-points for HDL, LDL, and cholesterol were 50, 130, and 210 mg/dL, respectively [42,43]. The cut-point for CRP was set to 3 mg/L [44]. We excluded participants with CRP more than 15 md/dL (*n* = 863) since it may be an indicator of acute infection. For insulin, HbA1c, and fasting glucose, we used 10.57 µU/mL, 6.5%, and 7.1 mmol/L as cut-points, respectively [45,46,47]. Values higher than the cut-points were considered the outcome of interest, except for HDL in which lower values were considered the outcome of interest in logistic regression models (see statistical analysis section).

### 2.3. Exposure Assessment

NHANES uses 24-h dietary recall interviews (24HRs). Data from both days of 24HR reporting were utilized in this work. NHANES processed the dietary data to obtain micro and macronutrients by using the USDA’s Food and Nutrient Database for Dietary Studies. To calculate fasting durations, average start and end times for consumption across the two days were utilized. If a participant only had the first day of 24HR reporting, then only the first day was used. Participants were asked to report every food they consumed from midnight to 11:59 p.m. For calculation of nighttime fasting duration, it is better to start with first consumption after waking and end with last consumption prior to sleep (although late night eating during the sleep period still needs to be considered). The NHANES diet reporting protocol required assumptions to be made for fasting duration derivation. The designation of “breakfast” was assumed to be the first meal of the day after waking. Therefore, any consumption of food or drinks prior to breakfast (i.e., between midnight and breakfast) was assumed to be the last consumption prior to bedtime for those going to sleep after midnight. As an illustration, suppose participant X consumed food at 1:00 a.m., then reported breakfast at 8:00 a.m., and ended their day of reporting with food consumption at 10:00 p.m. According to NHANES coding practices, the 1:00 a.m. food consumption was the first meal consumed during the day. However, it was most likely the last meal consumed prior to going to bed (on the previous day). Therefore, this participant would have a fasting time from 1:00 a.m. to 8:00 a.m., not 10:00 p.m. to 8:00 a.m. In referring to just the 2015–2016 cycle, only 18 participants had a dietary pattern where such an assumption had to be made. To calculate fasting durations, all food consumption times were converted to hours throughout a 24-h period. For example, food consumed at 8:30 a.m. was valued at 8.5 h. Foods consumed between midnight and breakfast were recoded to be later than 24 h (i.e., 1:00 a.m. was converted to 25 h). The algorithm to calculate fasting durations was: 24 − last consumption + first consumption. A participant that had breakfast at 8:00 a.m. and had their last consumption at 10:00 p.m. (i.e., 22 h) would have a fasting duration of 24 − 22 + 8 = 10 h of fasting. Fasting durations were analyzed continuously as well as in quartiles.

### 2.4. Statistical Analyses

All analyses accounted for the clustered sampling design and the examination survey weights as required for appropriate analysis of these data. Given the outcomes were available for a different number of 2-year cycles, various sampling weights were used. We used 6-year weights for CRP and 12-year weights for HbA1c %, and total and HDL cholesterol. We used 12-year fasting blood sample weights for fasting glucose, insulin, and LDL cholesterol. Population characteristics and crude outcome data were described by quartiles of fasting durations. For categorical measures, only weighted percentages were provided as raw sample sizes of weighted data are not very meaningful. For continuous measures, weighted means and standard deviations were presented.

We performed multiple linear regression models with potential confounders that may bias the relationship between fasting duration and the cardiometabolic markers. *a priori* selected covariates included age (per 10 years), sex (male or female), race/ethnicity (Mexican American, non-Hispanic white, non-Hispanic black, others), alcohol use (drinker, non-drinker), marital status (married/living with partner, window or divorced or separated, single), smoking status (current, former, never), sleep duration (<7 h, ≥7 h), use of cholesterol medication (yes or no), number of days measuring the fasting duration (one or two), and BMI (under or normal, overweight, obese), Energy-density Dietary Inflammatory Index (E-DII^®^, quartiles), and energy intake (quartiles). The E-DII quantifies the degree of anti- or pro-inflammatory potential [48] and served as a measure of diet quality for adjustment purposes. Results of regression analyses were presented as β-values (standard errors [SE]) and *p* values. We also explored quartiles of fasting durations with the primary comparison of interest occurring between quartiles 1 and 4. This approach allowed for quantification of least square means per fasting quartile which may increase interpretability and use in clinic settings. The timing of the last meal of the day and the timing of the first meal of the day were analyzed in the same exact manner as described above for fasting times. The quartile ranges for last mealtime and first mealtime can be found in Table 4. Upon reviewing results using first mealtime as the independent variable, it was decided to examine the interaction between first mealtime and fasting duration in attempts to explain the findings observed for fasting duration analyses. The quartile forms of first mealtime and fasting duration were used. Specifically, results using the first and fourth quartiles are presented as these represent the earliest and latest times for first mealtime and the shortest and longest fasting durations.

Serum biomarkers were then categorized using the clinical cut-points defined above. For these categorical outcomes, we conducted logistic regression to investigate the associations between the fasting duration and abnormality of serum biomarkers. All logistic models used the same adjustments as the linear models.

We performed several sensitivity analyses to test the robustness of our main results. First, we conducted analyses by excluding potential shift workers (*n* = 803). These were determined based on timing of main meals (i.e., breakfast, lunch, and dinner). For example, if a participant reported lunch at midnight and dinner at 4:00 a.m., it was assumed that such an individual was a shift worker. Second, we tested whether weekends or weekdays would affect our main results by excluding participants who reported their diets for weekend days (*n* = 451). Third, we repeated our analyses by excluding the extreme values of fasting duration, such as ≥23 or ≤2 h (*n* = 76). We set the two-sided *p* < 0.05 as statistically significant. All analyses were performed using SAS version 9.4 (SAS Institute Inc., Cary, NC, USA).

## 3. Results

Compared to those with the longest nighttime fasting duration, those with the shortest nighttime fasting duration were more likely to be currently married (55% vs. 67%), be non-Hispanic white (57% vs. 74%), consume alcohol (23% vs. 32%), be males (44% vs. 55%), and be older (mean age 43.9 vs. 48.3 years). All *p*-values for these differences were <0.01 (Table 1).

The primary analyses utilized fasting durations as a continuous metric. For every one-hour increase in nighttime fasting duration, average insulin and CRP increased by 0.287 uU/mL (standard error [SE] = 0.057, *p* < 0.01) and 0.03 mg/L (SE = 0.012, *p* = 0.02), respectively. Additionally, for every-hour increase in fasting duration, lower HDL of 0.101 mg/dL (SE = 0.046, *p* = 0.03) was observed. When using fasting durations as quartiles with the primary comparison between quartiles 4 and 1, those in quartile 4 (i.e., longest fasting duration) compared to quartile 1, had statistically significantly greater CRP (3.01 vs. 2.69 mg/dL, *p* < 0.001) and insulin (14.2 vs. 12.2 mg/dL, *p* < 0.001), and lower HDL cholesterol (53.6 vs. 54.3 mg/dL, *p* = 0.03, Table 2). These findings were opposite compared to the hypothesized direction.

Next, logistic regression was performed to examine odds of meeting clinical thresholds for each of the outcomes. Every one-hour increase in fasting duration was associated with a 7% (95%CI = 1.05–1.10) increase in odds of abnormal insulin and a 1% (95%CI = 1.00–1.03) increase in odds for abnormal HDL cholesterol. When comparing the highest quartile of fasting duration to the lower, a statistically significantly elevated odds for values beyond clinical thresholds was observed for CRP, insulin, and HDL cholesterol (Table 3). Again, these were in the opposite direction than hypothesized.

In attempts to understand results for fasting duration in the opposite direction than hypothesized, *post hoc* analyses examined last and first mealtimes. For last mealtimes, every one-hour increase in fasting was associated with a 0.873 (SE = 0.306, *p* = 0.01) lower value in LDL and a higher HbA1c % (β = 0.008, SE = 0.004, *p* = 0.05). Those in the last quartile (i.e., latest times) for last mealtime had higher levels of HbA1c % and insulin but lower values LDL and total cholesterol (all *p* < 0.05) compared to the first quartile. For the first mealtime, every one-hour increase was associated with an increase in CRP (β = 0.044, SE = 0.017, *p* = 0.0106), insulin (β = 0.429, SE = 0.105, *p* < 0.01), and glucose (β = 0.662, SE = 0.185, *p* < 0.01), and a decrease in HDL (β = −0.377, SE = 0.073, *p* < 0.01). Similarly, statistically significantly greater means in quartile 4 (latest first mealtime) compared to quartile 1 were observed for CRP, HbA1c %, insulin, and glucose, and lower HDL (Table 4).

To further explore the differences in results between fasting duration and first mealtime, results for first mealtime were stratified by fasting duration. Largely speaking, results for first mealtime among the total population were observed in quartile 1 and quartile 4 of fasting duration separately. For example, those with the latest first mealtime (i.e., quartile 4 of first mealtime) compared to the earliest first mealtime had statistically significant greater insulin among those with shorter fasting durations (14.0 vs. 11.9 uU/mL, *p* = 0.06) and in those with the longest fasting durations (14.7 vs. 10.4 uU/mL, *p* ≤ 0.01). Similar patterns of results (i.e., healthier values in the earliest mealtime regardless of fasting duration quartile) were observed for glucose. For HDL, a later first mealtime was associated with lower (i.e., worse) values compared to an earlier first mealtime, but only among those with shorter fasting durations (Table 5).

A few sensitivity analyses were performed for the results related to fasting duration in Table 2. First, we excluded those who were suspected of being shift workers, then those with extreme fasting durations (i.e., ≤2 or ≥23 h), and those who had a dietary report on the weekend (data not tabulated). With none of these did the interpretation of the findings change compared to results presented in Table 2.

## 4. Discussion

One concern with calorie restriction-based diet programs is weight regain [5,6]. Of growing interest, fasting and other chrononutrition-focused approaches (e.g., consuming a majority of calories early in the morning) have shown potential for improvements in cardiometabolic endpoints, as well as sustainability [37]. However, most of this work has been conducted in smaller scale intervention studies or pragmatic designs using Ramadan [26].

Therefore, this study examined the associations between nighttime fasting duration, first mealtime, and last mealtime with a range of cardiometabolic endpoints using representative data from NHANES. Two previous studies have examined meal timing-related constructs within NHANES. Previously, Kant and colleagues investigated the timing of breakfast, lunch, and dinner by sex and race in NHANES [49]. Marinac and colleagues found that for every 10% increase in calories consumed after 5:00 p.m. was associated with a 3% increase in CRP among women from a single two-year cycle of NHANES [50]. These studies were either descriptive in nature or only focused on women from a single two-year cycle of NHANES data.

Results of the current study related to fasting duration were mainly in the opposite direction than hypothesized. Increased fasting duration was associated with increased CRP and insulin, and decreased HDL. Additionally, positive point estimates were observed for the association between increased fasting duration and both LDL and total cholesterol although the estimates did not quite achieve statistical significance.

Studies using observational or pragmatic designs have shown associations between fasting and inflammatory markers [31,32]. For example, Faris and colleagues noted statistically and clinically significant reductions in interleukin (IL)-6, IL-1β, and tumor necrosis factor (TNF)-α at the end of Ramadan compared to prior to Ramadan [51]. Even though the current study did not use these inflammatory markers, such results conflict with the current findings. Interestingly, intervention studies tend to be inconsistent with no clear relationship between fasting and inflammatory markers [33,34]. The findings from the current study highlight the need to consider clinical importance, as opposed to statistical significance. Although longer fasting durations were statistically significantly associated with increased CRP, the beta coefficient was only 0.03 for a 1-h difference in fasting duration. Taken in context, a 10-h fasting difference between two individuals would equate to a 0.3 mg/L difference in CRP which is not biologically meaningful. Given this, results of the current study related to CRP are somewhat corroborated by past intervention studies.

Much of the work examining associations between metabolic endpoints and IF protocols tend to show beneficial effects. For example, one study employed a 10-h TRF protocol among participants with metabolic syndrome over 12 weeks and found fasting glucose decreased by 21% and HOMA-IR by 30% [34]. Jamshed and colleagues conducted an RCT where control participants consumed meals at 800, 1400, and 2000 h. The eTRF group consumed meals at 800, 1100, and 1400 h. In the eTRF group, 24-h mean glucose levels were reduced by 4 ± 1 mg/dL [52]. In the current study, every one-hour increase in fasting duration was associated with an increase in insulin by about 0.29 µU/mL which is opposite of what was expected.

As for cardiovascular markers, HDL was found to decrease with increased fasting duration and total and LDL cholesterol increased. However, again, results are not very clinically meaningful. This is consistent with previous research. A meta-analysis observed that the weighted mean difference between TRF and control groups for total cholesterol (2.43 mg/dL), HDL cholesterol (0.76 mg/dL), and LDL cholesterol (−0.44 mg/dL) were not statistically or clinically significant [26].

Although most results for fasting duration were not of clinical significance, the fairly consistent conclusion that results trended in a direction opposite of the hypothesis was perplexing. To further investigate this phenomenon, the timing of the last meal of the day and first meal of the day were analyzed. It was interesting to observe that later timing of the last meal was associated with higher HbA1c % and insulin which is somewhat supported by increased peripheral and hepatic insulin resistance observed towards the evening hours [29,30]. At the same time, LDL and total cholesterol were lower with later timing of the last meal. This is partially supported by work indicating that a greater meal frequency, which is often associated with consumption later into the evening, is associated with lower LDL and total cholesterol [53].

Every one-hour increase in the time of first meal was associated with higher CRP, HbA1c %, insulin, and glucose and lower HDL cholesterol. Previously, every one-hour increase in first meal consumption was associated with an increase in glycoprotein acetyl, a pro-inflammatory marker [54]. The phenomenon of a later time of first meal consumption may be analogous to breakfast skipping. Compared to always eating breakfast, those never eating breakfast had an increased odds of having a CRP value above 3.0 mg/L of 1.27 (95%CI = 1.15–1.40) [55]. In a review including 105 peer-reviewed publications, it was concluded that regular breakfast consumption, compared to regularly skipping, improved carbohydrate metabolism and reduced T2DM risk [56]. Interestingly, when results were stratified by fasting duration quartiles in the current study, there was no clear pattern in CRP values other than longer fasting durations having larger CRP values.

An earlier breakfast is also a key feature of eTRF. As noted above, Jamshed and colleagues observed reduced blood glucose with an eTRF protocol [52]. Another study that was a cross-over design among men with pre-diabetes randomized men to a control condition with meals at 700, 1300, and 1900 h and an intervention arm with meals at 700, 1000, and 1300 h. eTRF led to reduced fasting insulin by 3.4 ± 1.6 mU/L. Insulin levels were also reduced at 60 and 90 min post-load [57]. Starting consumption earlier in the day and ending earlier in the evening has been shown to have numerous cardiometabolic effects. Human studies have shown decreased insulin secretion and peripheral and hepatic insulin resistance in the evening compared to the morning [29,30], and later evening meals, compared to identical morning meals, result in hyperglycemic responses [58]. This could ultimately lead to increased systemic inflammation given that chronically high levels of glucose in the blood is a pro-inflammatory state, in and of itself [59]. When first mealtime results were stratified by fasting duration for the metabolic markers, results for first mealtime persisted across fasting duration quartiles. In other words, a later first mealtime, compared to earlier, was associated with increased (i.e., worse) metabolic parameters (i.e., insulin, glucose, and HbA1c %) regardless of fasting duration. Healthier (i.e., higher) HDL was observed among those with earlier first mealtimes, compared to later, but only among those with shorter fasting durations.

Strengths of this study include use of a large representative sample of the US population. Previous research by others has primarily relied on small highly controlled clinical trials or pragmatic designs involving Ramadan [36,37]. Such studies may be underpowered to test associations of interest or findings may not be generalizable to the general population. This is especially true of small clinical trials which often include relatively healthy participants from more affluent backgrounds. Most participants provided two days of 24HR which were averaged together to estimate fasting durations. This helps to address day-to-day variability in dietary reporting. NHANES uses rigorous biological sampling procedures. Lastly, a range of covariates were available for inclusion in models to account for a range of possible confounders.

However, this study was subject to a few limitations. One limitation may be the potential for residual confounding based on how the “nighttime fasting period” was defined. The 24HR started at midnight and went to 11:59 p.m. The difficulty with characterization of fasting durations given this diet collection protocol is that the sleep period is unknown and consecutive days of 24HR may not have been available. Hence, it was not possible to truly estimate the fasting period from one night to the next morning. Instead, last and first mealtimes had to be estimated which was difficult in situations where meals were consumed in the very early morning hours. However, a series of sensitivity analyses were conducted including removing those suspected of being shift workers, those with extreme fasting durations, and those with weekend dietary reports to try to account for potential biases related to how fasting was calculated. As with most cross-sectional studies, reverse causality is a concern. However, it is hard to conceive how abnormal inflammatory or metabolic markers could cause someone to change their fasting durations in the general population.

## 5. Conclusions

This study found that longer fasting durations were associated with higher CRP, insulin, and lower HDL cholesterol. These findings are in the opposite direction of what most of the literature has shown. However, upon further investigation, it was found that a later time of first meal consumption was associated with higher CRP, HbA1c %, insulin, glucose, and total and LDL cholesterol, and lower HDL cholesterol. Some of these findings resemble findings related to breakfast skipping. It is conceivable that through unhealthy eating practices (e.g., skipping breakfast, only eating one meal a day), that individuals artifactually create a longer fasting period which, in turn, is associated with abnormal biomarker levels. It would be interesting to conduct future studies among the general public using more rigorous meal timing data collection procedures to see if a similar phenomenon persists. Regardless, findings from this study tend to support evidence indicating that starting food consumption earlier in the day (e.g., eTRF) has beneficial cardiometabolic effects.

## Figures and Tables

**Table 1 nutrients-13-02686-t001:** Characteristics of participants according to the quartiles of fasting time in NHANES (2005–2016).

Characteristics	Quartiles of Fasting Duration			
	Quartile 1	Quartile 2	Quartile 3	Quartile 4
**Median of fasting duration (h)**	9.75	11.5	12.88	15
**Range of fasting duration (h)**	<10.75	10.75–12.16	12.17–13.75	≥13.75
**Smoking status (%)**				
Never	48	55	60	58
Former	27	26	24	20
Current	25	19	16	22
**Marital status (%)**				
Married/living with partner	67	68	65	55
Widow, divorced, separated	19	18	17	19
Single	14	15	18	26
**Race (%)**				
Mexican American	6	8	9	11
Non-Hispanic white	74	72	67	57
Non-Hispanic black	9	9	11	17
Other	11	12	13	14
**Alcohol drinker (Yes, %)**	32	30	26	23
**Cholesterol medication (Yes, %)**	17	17	18	14
**Gender (Female, %)**	44	52	57	55
**Age (years), mean (SE)**	48.3 (0.30)	47.7 (0.28)	47.8 (0.39)	43.9 (0.38)
**BMI (kg/m^2^), mean (SE)**	28.5 (0.12)	28.8 (0.12)	28.9 (0.11)	29.4 (0.13)
**Sleep duration (h), mean (SE)**	6.8 (0.02)	7.1 (0.02)	7.2 (0.02)	7.2 (0.02)
**Energy intake (kcal), mean (SE)**	2433 (19)	2243 (14)	2062 (13)	1897 (17)
**DII, mean (SE)**	0.41 (0.04)	0.28 (0.04)	0.34 (0.04)	0.73 (0.03)

Abbreviations: DII = Dietary Inflammation Index; MET = metabolic equivalents; SE = standard error; BMI = body mass index; MVPA = moderate-to-vigorous physical activity. Percentages, means, and standard errors (standard error) were estimated using 12-years US population weights. Categorical variables were presented as percentage and continuous variables were presented as mean (standard error). Column percentages may not equal 100% due to rounding. *p*-values were calculated from Chi-Square test for categorical variables and ANOVA for continuous variables. All *p*-values comparing the characteristics across fasting quartiles were ≤0.01.

**Table 2 nutrients-13-02686-t002:** Associations (least square means and standard errors) between fasting duration and cardiovascular biomarkers in NHANES (2005–2016).

Biomarker	Quartiles of Fasting Duration				Per 1 h β ± SE	Cont. *p*-Value
	Quartile 1	Quartile 2	Quartile 3	Quartile 4		
CRP, mg/dL	2.69 (0.08)	2.89 (0.09) *	2.94 (0.09) *	3.01 (0.07) *	0.030 (0.013)	0.02
Hb1Ac, %	5.85 (0.02)	5.86 (0.02)	5.89 (0.02)	5.85 (0.02)	−0.003 (0.003)	0.28
Insulin, uU/mL	12.2 (0.50)	13.0 (0.39) *	13.5 (0.42) *	14.2 (0.38) *	0.287 (0.057)	<0.01
Glucose, mg/dL	111.0 (0.90)	110.9 (0.73)	111.9 (0.75)	112.6 (0.91)	0.191 (0.147)	0.20
HDL, mg/dL	54.3 (0.28)	53.8 (0.31)	53.7 (0.29) *	53.6 (0.31) *	−0.103 (0.046)	0.03
LDL, mg/dL	106.1 (1.06)	106.0 (1.01)	107.1 (1.19)	106.8 (1.06)	0.287 (0.158)	0.07
Cholesterol, mg/dL	186.9 (0.77)	187.2 (0.87)	188.9 (0.80)	188.2 (0.74)	0.243 (0.130)	0.06

Abbreviations: CRP = C-reactive protein; HDL = high-density lipoproteins; LDL = low-density lipoproteins; SE = standard error; HbA1c = glycosylated hemoglobin. Models adjusted for age (per 10 years), race, gender, marital status, alcohol use, body mass index, smoking status, sleep duration, cholesterol medicine, days for dietary reports, energy intake, and the Dietary Inflammatory Index. Quartile ranges: Quartile 1: <10.75, Quartile 2: 10.75–12.16, Quartile 3: 12.17–13.74, Quartile 4: ≥10.75 * Represents means that were statistically significantly (*p* < 0.05) different compared to quartile 1.

**Table 3 nutrients-13-02686-t003:** Odds ratios and 95% confidence intervals for associations between fasting duration and abnormality of cardiovascular biomarkers in NHANES (2005–2016).

Biomarkers	Quartiles of Fasting Duration				
	Quartile 1	Quartile 2	Quartile 3	Quartile 4	Per 1 h
CRP ≥ 3 mg/dL	1.00 (Ref)	1.17 (1.00, 1.36)	1.20 (1.03, 1.40)	1.21 (1.05, 1.40)	1.01 (0.99, 1.04)
Hb1Ac ≥ 6.5%	1.00 (Ref)	1.06 (0.88, 1.26)	1.09 (0.93, 1.28)	1.01 (0.85, 1.21)	0.99 (0.97, 1.02)
Insulin ≥ 10.57 uU/mL	1.00 (Ref)	1.51 (1.31, 1.73)	1.62 (1.39, 1.89)	1.65 (1.37, 1.99)	1.07 (1.05, 1.10)
Glucose ≥ 126 mg/dL	1.00 (Ref)	1.00 (0.75, 1.34)	1.09 (0.87, 1.36)	1.13 (0.88, 1.44)	1.00 (0.97, 1.04)
HDL ≤ 50 mg/dL	1.00 (Ref)	1.05 (0.95, 1.16)	1.03 (0.94, 1.12)	1.10 (1.00, 1.20)	1.01 (1.00, 1.03)
LDL ≥ 130 mg/dL	1.00 (Ref)	1.03 (0.91, 1.17)	1.11 (0.97, 1.28)	1.10 (0.95, 1.28)	1.03 (1.00, 1.05)
Cholesterol ≥ 210 mg/dL	1.00 (Ref)	0.97 (0.89, 1.07)	1.07 (0.96, 1.19)	1.05 (0.92, 1.18)	1.01 (1.00, 1.03)

Abbreviations: CRP = C-reactive protein; HDL = high-density lipoproteins; LDL = low-density lipoproteins; HbA1c = glycosylated hemoglobin; Ref = reference. Model adjusted for age (per 10 years), race, gender, marital status, alcohol use, body mass index, smoking status, sleep duration, cholesterol medicine, days for dietary reports, energy intake, and the Dietary Inflammatory Index. Quartile ranges: Quartile 1: <10.75, Quartile 2: 10.75–12.16, Quartile 3: 12.17–13.74, Quartile 4: ≥10.75.

**Table 4 nutrients-13-02686-t004:** Associations (least square means and standard errors) between time of last meal and first meal and cardiometabolic biomarkers in NHANES (2005–2016).

Last Mealtime						
Biomarker	Quartiles of Time of Last Meal				Per 1 h β ± SE	Cont. *p*-Value
	Quartile 1	Quartile 2	Quartile 3	Quartile 4		
CRP, mg/dL	2.84 (0.08)	2.86 (0.10)	2.87 (0.08)	2.89 (0.09)	−0.002 (0.022)	0.94
Hb1Ac, %	5.83 (0.02)	5.85 (0.02)	5.87 (0.02)	5.87 (0.02) *	0.009 (0.005)	0.05
Insulin, uU/mL	12.5 (0.46)	13.3 (0.52) *	13.7 (0.36) *	13.4 (0.39) *	0.040 (0.082)	0.63
Glucose, mg/dL	110.9 (0.85)	110.8 (0.81)	112.2 (0.81)	112.1(0.88)	0.371 (0.223)	0.10
HDL, mg/dL	53.9 (0.35)	54.3 (0.33)	53.9 (0.32)	53.7 (0.31)	−0.104 (0.070)	0.14
LDL, mg/dL	108.2 (1.48)	107.0 (1.45)	107.2 (1.01)	104.1 (1.12) *	−0.873 (0.306)	0.01
Cholesterol, mg/dL	188.5 (0.96)	188.0 (0.86)	188.2 (0.74)	186.4 (0.75) *	−0.378 (0.224)	0.10
**First Mealtime**						
**Biomarker**	**Quartiles of Time of First Meal**				**Per 1 h** **β ± SE**	**Cont.** ***p*-Value**
	**Quartile 1**	**Quartile 2**	**Quartile 3**	**Quartile 4**		
CRP, mg/dL	2.72 (0.09)	2.86 (0.10)	2.95 (0.08) *	2.96 (0.09) *	0.044 (0.017)	0.01
Hb1Ac, %	5.84 (0.02)	5.82 (0.02)	5.87 (0.02) *	5.90 (0.02) *	0.007 (0.004)	0.06
Insulin, uU/mL	11.9 (0.51)	12.7 (0.46) *	13.2 (0.43) *	14.8 (0.35) *	0.429 (0.105)	<0.01
Glucose, mg/dL	109.7 (0.84)	109.7 (0.82)	111.9 (0.71) *	114.5 (1.16) *	0.662 (0.185)	<0.01
HDL, mg/dL	54.7 (0.30)	54.4 (0.38)	53.1 (0.29) *	53.1 (0.34) *	−0.377 (0.073)	<0.01
LDL, mg/dL	106.3 (0.91)	106.7 (1.12)	107.3 (1.26)	104.6 (1.18)	−0.361 (0.209)	0.09
Cholesterol, mg/dL	187.0 (0.67)	188.9 (0.82) *	187.3 (0.87)	185.9 (0.80)	−0.302 (0.154)	0.05

Abbreviations: CRP = C-reactive protein; DII = Dietary Inflammation Index; HDL = high-density lipoproteins; LDL = low-density lipoproteins; HbA1c = glycosylated hemoglobin. Model adjusted for age (per 10 years), race, gender, marital status, alcohol use, body mass index, smoking status, sleep duration, cholesterol medicine, days for dietary reports, and the Dietary Inflammatory Index. The last meal time was additionally adjusted for total energy intake between 3:00–8:00 p.m. and 8:00+ p.m. First meal intake was additionally adjusted for total energy intake between 4:00–10:00 a.m. and 10:00 a.m.–3 p.m. First mealtime quartile ranges: Quartile 1: <7:23 a.m., Quartile 2: 7:23–8:30 a.m., Quartile 3: 8:30–9:45 a.m., Quartile 4: ≥9:46 a.m. Last mealtime quartile ranges: Quartile 1: <7:15PM, Quartile 2: 7:15–8:15 p.m., Quartile 3: 8:16–9:30 p.m., Quartile 4: ≥9:30 p.m. * Represents means that were statistically significantly (*p* < 0.05) different compared to quartile 1.

**Table 5 nutrients-13-02686-t005:** Least square means and standard errors of outcomes by quartiles 1 and 4 of first mealtime and fasting duration in NHANES (2005–2016).

Quartiles of First Mealtime	Quartiles of Fasting Duration		Interaction *p*-Value
	Quartile 1	Quartile 4	
CRP, mg/dL			0.10
Quartile 1	2.60 (0.10)	3.43 (0.35)	
Quartile 4	2.65 (0.21)	3.08 (0.09)	
Hb1Ac, %			0.04
Quartile 1	5.86 (0.02)	5.80 (0.08)	
Quartile 4	5.87 (0.05)	5.89 (0.02)	
Insulin, uU/mL			0.72
Quartile 1	11.9 (0.64)	10.4 (0.74)	
Quartile 4	14.0 (1.19)	14.7 (0.4) *	
Glucose, mg/dL			0.92
Quartile 1	110.3 (1.11)	108.57 (2.6)	
Quartile 4	113.6 (2.86)	114.1 (1.41) *	
HDL, mg/dL			0.19
Quartile 1	54.9 (0.34)	52.5 (1.21)	
Quartile 4	52.3 (1.03) *	53.2 (0.04)	
LDL, mg/dL			0.66
Quartile 1	106.3 (1.10)	110.7 (4.50)	
Quartile 4	102.4 (3.65)	105.3 (1.26)	
Cholesterol, mg/dL			0.48
Quartile 1	186.9 (0.78)	188.8 (3.36)	
Quartile 4	181.9 (2.25) *	186.0 (0.93)	

Abbreviations: CRP = C-reactive protein; HDL = high-density lipoproteins; LDL = low-density lipoproteins; SE = standard error; HbA1c = glycosylated hemoglobin. Models adjusted for age (per 10 years), race, gender, marital status, alcohol use, body mass index, smoking status, sleep duration, cholesterol medicine, days for dietary reports, energy intake, and the Dietary Inflammatory Index and total energy intake between 4 a.m.–10 a.m. and 10 a.m.–3 p.m. Interaction *p*-values represent the *p*-values for the multiplicative interaction between first mealtime and fasting duration quartiles. * Represents means that were statistically significantly (*p* < 0.05) different between fasting quartiles within quartile 1 or 4 of fasting duration.

## Data Availability

Data available in a publicly accessible repository. Publicly available datasets were analyzed in this study. This data can be found here: [https://www.cdc.gov/nchs/nhanes/index.htm], accessed on 2 August 2021.

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
