# Peer review of "Associations between Fasting Duration, Timing of First and Last Meal, and Cardiometabolic Endpoints in the National Health and Nutrition Examination Survey"

_nutrients, 2021, doi:10.3390/nu13082686_

Round 1

Reviewer 1 Report

This author acquired and analyzed data from National Health and Nutrition Examination Survey (NHANES) to investigate the effects of fasting duration, timing of first and last Meal, and plasma cardiometabolic profiles in US populations. The results are interesting and may pinpoint important interventions for the better diet regimen. However there are many points needed to be addressed. 

  1. Abstract: should be in an one paragraph style without subtitles.
  2. Introduction section: contains a lot of literature review. Introduction needs to be reconstructed into more paragraphs with one or same ideas per paragraph.  
  3. Author already stated the objective of this study in the last paragraph. However, it is too long with a lot of citing backgrounds. Separate and reconstruct the Introduction section more clearly. Separate and only state the purpose of this study in the last paragraph of Introduction
  4. Methods: Do you need consent for data from NHANES? State the consent document number if required. 
  5. Results section: Where is the description related to Table 1? I don't see it between Line 221-272
  6. In Table 1, it is required to perform the statistics in each indicator among different quartiles and "mark the significance by * or by English letters". Showing P values is even better.
  7. In Table 4, what's the criteria to divide human subjects into different quartiles for "First Meal Time" or "Last Meal Time" among different quartiles 1, 2, 3, 4? The methods need to be described in Method section and Table 4.
  8. Discussion section: each paragraph is too long. Needs to be reconstructed into more paragraphs with one or same ideas per paragraph.  

Author Response

Reviewer 1

Comment 1: Abstract: should be in an one paragraph style without subtitles.

Response: Thank you for pointing this out to us. The journal already took care of this for us as we submitted the manuscript using the “non-formatted” submission option through Nutrients. They then take the unformatted manuscript and format on behalf of the authors. The Abstract now appears as a single paragraph.

Comment 2: Introduction section: contains a lot of literature review. Introduction needs to be reconstructed into more paragraphs with one or same ideas per paragraph.  

Response: We are not entirely clear what sections of the Introduction should be split into more paragraphs. As a reviewer, you will have access to the other reviewer’s comments. As you will see from those comments, the other reviewer is suggesting that more literature should be added to the background. So, we have received conflicting comments from the two reviewers. We did our best to address both reviewers’ concerns but had to make some compromises. We split a couple of paragraphs but had to add more text for the other reviewer. Please see the Introduction for restructuring of paragraphs.

Comment 3: Author already stated the objective of this study in the last paragraph. However, it is too long with a lot of citing backgrounds. Separate and reconstruct the Introduction section more clearly. Separate and only state the purpose of this study in the last paragraph of Introduction

Response: In editing the Introduction to address Comment 2, we made this change as well. Please see page 3, lines 110-112 for the new purpose statement paragraph.

Comment 4: Methods: Do you need consent for data from NHANES? State the consent document number if required. 

Response: There is no specific consent document number that we can refer to. However, we did state that participants signed informed consent. Please see page 3, lines 129-131, for these details.

Comment 5: Results section: Where is the description related to Table 1? I don't see it between Line 221-272

Response: Thank you for pointing this out; that was a great catch. We neglected to identify results related to Table 1 in the Results section. See the reference to Table one inserted on page 5, line 252.

Comment 6: In Table 1, it is required to perform the statistics in each indicator among different quartiles and "mark the significance by * or by English letters". Showing P values is even better.

Response: Yes, we agree that traditionally this is the best approach. However, with NHANES data, weighting should be used. The weighted sample size of adults in NHANES totals over 100,000,000. Therefore, even minute differences in population characteristics by fasting quartiles will be highly statistically significant. Therefore, it is relatively meaningless to show the p-values. The footnotes of Table 1 do include a note indicating that chi-square or ANOVAs were run for all population characteristics in Table 1. The footnote also indicates that all were statistically significant at a p < 0.01.

Comment 7: In Table 4, what's the criteria to divide human subjects into different quartiles for "First Meal Time" or "Last Meal Time" among different quartiles 1, 2, 3, 4? The methods need to be described in Method section and Table 4.

Response: Again, thank you for pointing this out. We, in fact, did not describe the quartiles for these in the paper. We have included the ranges in the footnotes of Table 4. We also included a sentence in the Methods to direct readers to review Table 4 footnotes for the quartile ranges (see page 5, lines 223-224).

Comment 8: Discussion section: each paragraph is too long. Needs to be reconstructed into more paragraphs with one or same ideas per paragraph.  

Response: As with Comment 2, we had to add more to the Discussion text, as per Reviewer 2’s comments. With the Discussion, we tried to compare and contrast our results to past studies. Splitting up the paragraphs too much may negatively impact the flow of the Discussion. However, we found a few places where we were able to split up a paragraph. Please see changes throughout the Discussion for restructuring of the paragraphs.

Reviewer 2 Report

The authors of the manuscript "Associations between Fasting Duration, Timing of First and Last Meal, and Cardiometabolic Endpoints in the National Health and Nutrition Examination Survey" provide interesting, deep insights into chrono-nutritional data of the NHANES Study. Perplexingly, they find conflicting data suggesting that longer fasting durations do not, or even worsen, cardiometabolic parameters when the position of the eating window throughout the day is ignored. Additional analyses by the authors, however, suggest a superior importance of starting the first meal early in the day, which is kind of in line with a growing body of evidence supporting early TRE over late TRE.

The manuscript is well written and of high interest to the readership of the journal. I only have 1 major point for the authors' and editors' consideration and a few minor points that need to be addressed :

Major:

  • Based on the 2 present types of analyses in the manuscript, it should be possible to perform a third analysis to compare the differences in sub-populations that undergo the same fasting-duration, but start eating at different times in the day (e.g. compare people in quartiles 1 and 4 of "first meal time" with matched fasting-durations - focusing on those with longer fasting windows). This is an obvious next step to do, as it would approximately mirror the design of several studies that compare early vs. late TRE. Also, this would immensely strengthen the manuscript and its citability, independent of the analysis' outcome.

Minor:

  • Lines 54-57 seem out of place and do not add value to the introduction.
  • The body of literature cited in the manuscript is lacking a great deal of important studies by Mattison, deCabo, Longo, Stekovic, Weiss, the CALERIE consortium and so on. There have been several recent publications on the topic that need to be given credit.
  • There are several studies matching the caloric restriction of TRE/TRF groups to continuous CR. This needs to be properly discussed in the manuscript.
  • Ref 37: date of access is missing
  • Please indicate the fasting duration of the quartiles in tables 2-3, not only in table 1. This helps readers to grasp the data.
  • Please explain why both linear regression models and cubic splines are used.
  • Indicate what times the quartiles of first/last meals correspond to.
  • Lines 281/282: Over-simplifying. There are many well-controlled TRF studies that do not use Ramadan-practicing cohorts.

Author Response

Reviewer 2:

Comment 1: Based on the 2 present types of analyses in the manuscript, it should be possible to perform a third analysis to compare the differences in sub-populations that undergo the same fasting-duration, but start eating at different times in the day (e.g. compare people in quartiles 1 and 4 of "first meal time" with matched fasting-durations - focusing on those with longer fasting windows). This is an obvious next step to do, as it would approximately mirror the design of several studies that compare early vs. late TRE. Also, this would immensely strengthen the manuscript and its citability, independent of the analysis' outcome.

Response: Excellent suggestion! After seeing the results, we agree, this does strengthen the paper and adds to the importance of first mealtime findings. Overall, we found that an earlier first mealtime was associated with improved metabolic biomarker levels regardless of fasting duration. The same can be said for HDL. We added this new analysis to the Methods (see page 5, lines 224-229). The results can be found at page 6, lines 294-304. We added to the Discussion in a couple of locations related to these results (see page 12, lines 399-401, and page 12, lines 415-420).

Comment 2: Lines 54-57 seem out of place and do not add value to the introduction.

Response: We removed the lines in question.

Comment 3: The body of literature cited in the manuscript is lacking a great deal of important studies by Mattison, deCabo, Longo, Stekovic, Weiss, the CALERIE consortium and so on. There have been several recent publications on the topic that need to be given credit.

Response: Thank you for pointing out the presence of other important work that should be noted. However, we are not entirely clear as to which work you are referring to. There is a fair bit of work by the list of authors you have noted. We did a search on the names with inclusion of the term “fasting” and did find a few papers. If what you were referring to was not what we included, please provide more specific details as to which papers or topics specifically you were referring. We would be happy to evaluate them for their fit in this manuscript. For changes made based on this comment, see page 2, lines 59-63 for inclusion of a review by de Cabo and Mattson. We also were able to modify the text in a way to incorporate two citations that included Longo as an author which were appropriate for the text. We added a reference from Stekovic as well (see page 2, lines 84-86).

Comment 4: There are several studies matching the caloric restriction of TRE/TRF groups to continuous CR. This needs to be properly discussed in the manuscript.

Response: We kindly ask the reviewer to be more specific in their suggestion. We had a hard time understanding what is being asked of us and, therefore, we had a hard time finding appropriate evidence to include. The reviewer indicated studies “matching caloric restriction of TRE/TRF groups to continuous CR” should be discussed. This is the point we are making in the Introduction on page 2, lines 89-91. If studies match caloric restriction in TRF to continuous CR, then how can one truly examine the impact of TRF if results are similar between TRF and continuous CR? A review by Rynders indicated that 9 of 11 studies of various fasting protocols compared to CR led to the same weight reduction. However, in most of those studies, the calorie restriction in the TRF groups was similar to that of the continuous energy restriction group. It is hard to argue if the effects in the fasting groups were due to fasting or to moderate calorie restriction. What is needed are studies in which baseline energy intake is held constant within the fasting group compared to calorie restriction. Then, as a science, we can begin to parse out the effect of fasting vs. calorie restriction. We attempted to address this concern to explain this better by discussing the review that we refer to in this comment (see page 2, lines 91-94). If the reviewer has specific references to point us to, we are very open to reviewing them for their potential inclusion in the paper.

Comment 5: Ref 37: date of access is missing

Response: We removed this reference and replaced it with a different one that does not rely on the use of a website for referencing.

Comment 6: Please indicate the fasting duration of the quartiles in tables 2-3, not only in table 1. This helps readers to grasp the data.

Response: Yes, we agree that this should have been done to help Tables 2 and 3 serve as stand-alone tables. We have inserted the fasting duration quartiles into the footnotes of Tables 2 and 3. Additionally, we neglected to put the quartiles of first meal and last mealtime into Table 4. We have added those as well. Please refer to the footnotes of Table 4 for those.

Comment 7: Please explain why both linear regression models and cubic splines are used.

Response: We decided to remove the references to cubic splines. This was an approach to better understand the acceptance or violation of assumptions of linear regression. The assumptions of linear regression were upheld. Hence, there is no real need to discuss the cubic spline results.

Comment 8: Indicate what times the quartiles of first/last meals correspond to.

Response: We added the ranges for quartiles of first and last mealtime to the footnotes of Table 4.

Comment 9: Lines 281/282: Over-simplifying. There are many well-controlled TRF studies that do not use Ramadan-practicing cohorts.

Response: We agree with the statement from the reviewer that there have been other studies of TRF that did not involve Ramadan. We discussed some of those studies and mentioned some limitations of those previous studies. However, we felt we did need to highlight the limitations of small clinical studies more. We added a sentence about that on page 12, lines 425-427.

Round 2

Reviewer 1 Report

Authors answered most questions. However, there are still some unanswered questions. 

For example, previous comment 7: In Table 4, what's the criteria to divide human subjects into different quartiles for "First Meal Time" or "Last Meal Time" among different quartiles 1, 2, 3, 4? The methods need to be described in Method section and Table 4.

Author's response: Again, thank you for pointing this out. We, in fact, did not describe the quartiles for these in the paper. We have included the ranges in the footnotes of Table 4. We also included a sentence in the Methods to direct readers to review Table 4 footnotes for the quartile ranges (see page 5, lines 223-224).

Revision Question:

Author described "The quartiles ranges for last meal time and first meal time can be found 223 in Table 4" 
However, we can not see quartiles ranges. Author described quartile ranges in Table 2 and 3. In Table 4, quartiles ranges are still missing. 

Author Response

Reviewer 1

Comment 1: Author described "The quartiles ranges for last meal time and first meal time can be found in Table 4". However, we can not see quartiles ranges. Author described quartile ranges in Table 2 and 3. In Table 4, quartiles ranges are still missing. 

Response: We apologize for this oversight. We added the quartiles to the footnotes of Table 4.

Reviewer 2 Report

The edits have definitely improved the manuscript and it will be a valuable and highly citable addition to the literature on TRE. I have only minor suggestions left:

  • The new results description starting at line 294 does not have a reference to table 5. That should be included.
  • Line 417: "In other words, a later first meal time, compared to earlier, was associated with metabolic parameters (i.e., insulin, glucose, and HbA1c %) regardless of fasting duration." I think a describing adjective is missing before "metabolic parameters". Otherwise, I don't understand the sentence, since an association can mean anything. Can you clarify what you mean?
  • Please consider discussing the following articles:
    • https://stm.sciencemag.org/content/13/598/eabd8034
    • https://jamanetwork.com/journals/jamainternalmedicine/article-abstract/2771095

Author Response

Reviewer 2

Comment 1: The new results description starting at line 294 does not have a reference to table 5. That should be included.

Response: Thank you for catching this oversight. We’ve added a reference to Table 5 (see page 6, line 289).

Comment 2: Line 417: "In other words, a later first meal time, compared to earlier, was associated with metabolic parameters (i.e., insulin, glucose, and HbA1c %) regardless of fasting duration." I think a describing adjective is missing before "metabolic parameters". Otherwise, I don't understand the sentence, since an association can mean anything. Can you clarify what you mean?

Response: We completely see what the reviewer is referring to. We reworded the sentence to make it clearer that a later first mealtime was associated with worse (i.e., higher) values for those markers. In addition to this, we needed to revise our interpretation of HDL. See the new text in the Results (page 6, lines 286-288) and in the Discussion (page 12, lines 403-408).

Comment 3: Please consider discussing the following articles:

https://stm.sciencemag.org/content/13/598/eabd8034

https://jamanetwork.com/journals/jamainternalmedicine/article-abstract/2771095

Response: Thank you for providing links to specific papers to consider. The first one listed by Templeman and colleagues is very interesting. We missed that paper in our search. We have added that to the paper (see page 2, lines 90-94). As for the paper by Lowe, we carefully evaluated the paper but decided not to include it. In that paper, they noted “Although we did not measure calorie intake, mathematical modeling of changes in energy intake suggests that calorie intake did not significantly differ between groups.” This could very well indicate that the TRF group reduced calories in a manner similar to the calorie restriction group. Hence, there is no difference in calorie intake. This fits with the text on lines 85-89 for which we already have a review paper cited. We decided not to include the paper by Lowe for this reason.